# Coherent electrical readout of defect spins in silicon carbide by photo-ionization at ambient conditions

Matthias Niethammer[1]*, Matthias Widmann[1], Torsten Rendler[1], Naoya Morioka [1], Yu-Chen Chen[1], Rainer Stöhr [1], Jawad Ul Hassan [2], Shinobu Onoda [3], Takeshi Ohshima [3], Sang-Yun Lee [4], Amlan Mukherjee[1], Junichi Isoya [5], Nguyen Tien Son [2] & Jörg Wrachtrup[1,6]

Quantum technology relies on proper hardware, enabling coherent quantum state control as well as efficient quantum state readout. In this regard, wide-bandgap semiconductors are an emerging material platform with scalable wafer fabrication methods, hosting several promising spin-active point defects. Conventional readout protocols for defect spins rely on fluorescence detection and are limited by a low photon collection efficiency. Here, we demonstrate a photo-electrical detection technique for electron spins of silicon vacancy ensembles in the 4H polytype of silicon carbide (SiC). Further, we show coherent spin state control, proving that this electrical readout technique enables detection of coherent spin motion. Our readout works at ambient conditions, while other electrical readout approaches are often limited to low temperatures or high magnetic fields. Considering the excellent maturity of SiC electronics with the outstanding coherence properties of SiC defects, the approach presented here holds promises for scalability of future SiC quantum devices.

[1] 3rd Institute of Physics and Center for Applied Quantum Technologies, University of Stuttgart, 70569 Stuttgart, Germany. [2] Department of Physics, Chemistry and Biology, Linköping University, SE-581 83 Linköping, Sweden. [3] National Institutes for Quantum and Radiological Science and Technology, Takasaki 370-1292, Japan. [4] Center for Quantum Information, Korea Institute of Science and Technology, Seoul 02792, Republic of Korea. [5] Faculty of Pure and Applied Sciences, University of Tsukuba, Tsukuba 305-8573, Japan. [6] Max Planck Institute for Solid State Research, 70569 Stuttgart, Germany. *email: matthias.niethammer@pi3.uni-stuttgart.de

Solid state color centers have developed into a leading contender in quantum technology owing to their vast potential as hardware for quantum sensing and quantum networks[1-6]. Typically, these techniques employ optical control for spin state initialization and readout. Spins in solids can provide long spin relaxation and dephasing times and therefore constitute excellent quantum bits. In certain cases, e.g., for spins in wide-bandgap semiconductors, single spin manipulation and optical spin state readout is feasible[7-10]. A number of systems, like spin dopants in silicon[11-13] or quantum dots[14,15], allow for electrical spin readout. However, because their spin polarization typically relies on Boltzmann statistics, they require low temperature operation or large magnetic fields[16].

In contrast, color centers in wide-bandgap semiconductors show efficient optical spin polarization at room temperature[17,18]. Electrical readout of color center spins at ambient conditions relies on an efficient mechanism for spin-to-current conversion. This can be realized by measuring a laser induced spin-dependent photocurrent, which is often referred to as photocurrent detected magnetic resonance (PDMR). Several publications have successfully demonstrated this principle for various materials[19-22]. Recently, this technique has been applied to the nitrogen-vacancy (NV) center in diamond, by combining electrical readout with optical excitation[23,24] and even achieved single defect[25] sensitivity. It turns out that the signal-to-noise ratio (SNR) in this approach can outperform optical detection[25] and at the same time allows better integration into electronic periphery. However, diamond as host material is not compatible with industrial technologies, e.g., large-scale wafers and the development of efficient diamond electronics is still subject to research. Silicon carbide (SiC) on the other hand has attracted attention due to its outstanding optical, electrical and mechanical properties[3].

Traditionally, interest in defects in SiC was driven by their impeding properties to high power electronic devices[26]. This has initialized a wealth of studies utilizing electron paramagnetic resonance[27,28] and electrically detected magnetic resonance[29-33]. Among many investigated phenomena, spin dependent recombination has been shown to allow for self-calibrating magnetometers in a non-coherent fashion[34]. In addition, several spin-active defects with long spin coherence times[10,35] even at room temperature[9,36] have been found and their controlled and scalable fabrication is an active field of research[37-40]. The quantum properties of such color centers have lately been used to demonstrate magnetic field and temperature sensing[41-45].

In this work, we demonstrate electrical readout of a negatively charged silicon vacancy ($V_{Si}^{-}$) spin ensemble in a 4H-SiC device via PDMR at ambient conditions.

## Results

**Principle of PDMR of $V_{Si}^{-}$ (V2) in 4H-SiC.** First we want to introduce the $V_{Si}^{-}$, which will later be detected by PDMR. The negatively charged silicon vacancy $V_{Si}^{-}$ at the cubic lattice site (V2) in 4H-SiC provides both, stable deep level energy states in a wide-bandgap host and a spin dependent intersystem crossing (ISC). Previous studies revealed, that the defect has a spin quartet manifold of S = 3/2[28,46] in ground state (GS) and excited state (ES), which are separated by 1.35 eV (916 nm)[47]. GS and ES Landé g-factors are identical (g = 2.003) and their respective zero field splittings (ZFS) are 70 MHz and ≈410 MHz[48] at ambient conditions. In addition, a long-lived metastable state (MS) gives rise to non-radiative and spin-dependent ISC relaxation, enabling optical spin state initialization and readout under ambient conditions[9,47,49]. Furthermore, it provides excellent coherence times even at room-temperature[9,36,50]. However, small contrast in optically detected magnetic resonance (ODMR), low photon count rate, limited photon collection efficiency, and the lack of high quantum efficiency near-infrared detectors result in long integration times. Thus, new detection methods that can circumvent these limitations in the future are highly desired.

In the following, we discuss the principle of PDMR and how it can be applied to $V_{Si}^{-}$. Figs. 1a–c depict the underlying charge dynamics: In Fig. 1a, a deep level defect absorbs a photon and is excited from its GS to the ES. From there: (i) The system can decay back to the GS by emitting a photon. (ii) The system can undergo a non-radiative ISC via a metastable state. (iii) While being in the ES, the system can undergo a second optical excitation to the conduction band (CB) as shown in Fig. 1b. While case (i) and (ii) form the basis for ODMR, case (iii) is crucial for PDMR and will be discussed further. Hereby, an excess electron populates the CB, and the defect charge state $n$ is changed to $n + 1$. Now the defect can accept an electron in order to return the system to its equilibrium charge distribution. For this, multiple mechanisms are available: (A) The defect can recapture the excess electron from the CB, resulting in no net current. (B) Charge-transfer from other traps or defects can recharge the defects. (C) Electron capture from the valence band (VB) through photo-induced hole generation (Fig. 1c). While option (A) cannot contribute to the PDMR signal as the overall current has not been changed, options (B) and (C) remain of interest. As (B) involves complex mechanisms that strongly depend on band-bending, defect densities, excitation wavelength, and temperature[51,52], it can change the free charge carrier density and by this contribute to the PDMR. As the process (C) can be assumed to be fast, we expect this to be a major contributing mechanism in recharging[53]. The charge cycle is now complete

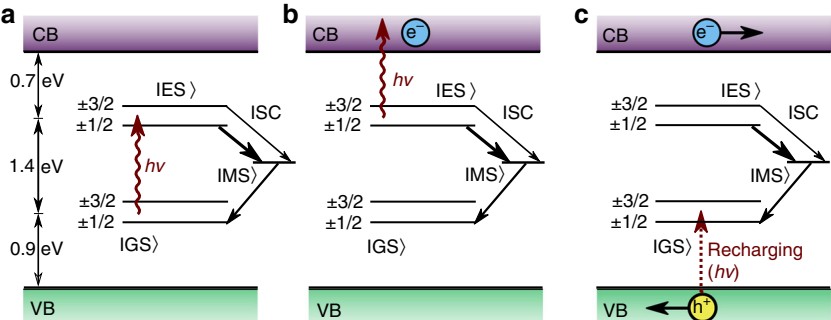

**Fig. 1** Photocurrent detected magnetic resonance (PDMR) mechanism and readout. **a** Single photon excitation. Excited state (ES) can relax to either ground state (GS) or metastable state (MS). Intersystem crossing (ISC) to MS is dependent on the spin state in ES, thus GS is polarized. **b** Second photon ionizes the defect and introduces a free electron in the conduction band (CB). **c** Recharging of the defect from valence band (VB) and separation of charges lead to a photocurrent.

and as shown in Fig. 1c the free electron-hole pair created in the process can be detected as photocurrent. If the ISC rates are spin-dependent, a change in spin state leads to different probabilities for the non-radiative relaxation and the photo-ionization path. This enables photo-electrical spin-state readout. As the energy difference from MS to CB is not known yet, we will take into account both MS and ES as photo-ionization sources in the following. The ratio of time spent in the ES and MS depends on the ISC rate. As the lifetime of MS is longer than of the ES, it is expected to see a larger cycling rate when the MS is not populated. We assume the $V_{Si}^-$ to be initialized in the $\pm 1/2$ spin subspace of the GS by optical illumination. During optical excitation, the ES is populated. If the ISC rate from ES to MS states is larger for $\pm 1/2$ than for the $\pm 3/2$ states, the chance for two-photon absorption of $\pm 3/2$ states from ES is higher, as the MS is less often populated. Contrary, the $\pm 1/2$ states will more likely decay to the MS and the cycling rate will be smaller. Thus we expect an increase in photocurrent from the ES when driving the spin from the $\pm 1/2$ subset to the $\pm 3/2$ states in the GS.

For an ionization from the MS to CB, a decrease in photocurrent should be measured when the spin system is resonantly driven to $\pm 3/2$. The overall sign and magnitude of the effect will thus be determined by the more dominant process and depends on the difference in absorption cross section, ISC rates, lifetime, and population of the ES and MS.

**PDMR device structure.** Structures for photocurrent detection usually consist of Schottky type, p–i–n, or p–n diodes. Such devices are prone to radiofrequency (RF) rectification. To minimize this effect, we choose a $n^{++}/n^-/n^{++}$ metal-semiconductor-metal junction, which is shown in Fig. 2a. Starting from a n-type 4H-SiC substrate, epitaxial growth was used to fabricate a three-layer stack: (i) a 10 μm-thick vanadium-doped semi-insulating

layer to reduce leakage currents into the substrate, (ii) a 10 μm-thick $n^-$ layer with N-doping concentration of $1 \times 10^{14}$ cm$^{-3}$, and (iii) a 400 nm-thick $n^{++}$ layer with N-doping concentration of $8 \times 10^{17}$ cm$^{-3}$. A nickel (Ni) layer of 100 nm thickness was deposited forming a Schottky contact on the $n^{++}$ layer. The sample was etched down by 10 μm, leaving fingers of various width as devices. Subsequently, the Ni and $n^{++}$ films were removed in rectangular center areas of various sizes, for optical access to the $n^-$ region (see zoomed inset in Fig. 2a). In this layer, we expect the charge state of the $V_{Si}^-$ to be stable. In addition, gold is deposited on the contact pads for wire bonding (see Supplementary Method 2).

After recording $I–V$ curves of the device, we create a $V_{Si}^-$ ensemble by electron irradiation at 2 MeV with a dose of $1 \times 10^{17}$ cm$^{-2}$. This process degrades the contact quality and device conductivity due to carrier compensation[54] (see Supplementary Note 1). However, this also results in minimizing the leakage current, enabling us to maximize the amplifier gain, which is beneficial for electrical readout. We chose to perform measurements on a device with 10 μm × 12 μm active area.

**Experimental implementation of PDMR.** Optical excitation of $V_{Si}^-$ defects is performed with a 785 nm laser. A 3D Helmholtz coil arrangement is used for applying magnetic fields in arbitrary directions. RF signals for spin control and manipulation are provided by a signal generator and applied via a coplanar waveguide on the printed circuit-board sample holder below the sample. This sample holder also incorporates contact pads, to which the device contacts are wire-bonded. For better SNR, we use a lock-in detection scheme. Therefore the signal is locked to the laser pulses for photocurrent measurements and onto the modulated RF pulses for ODMR, PDMR, and Rabi

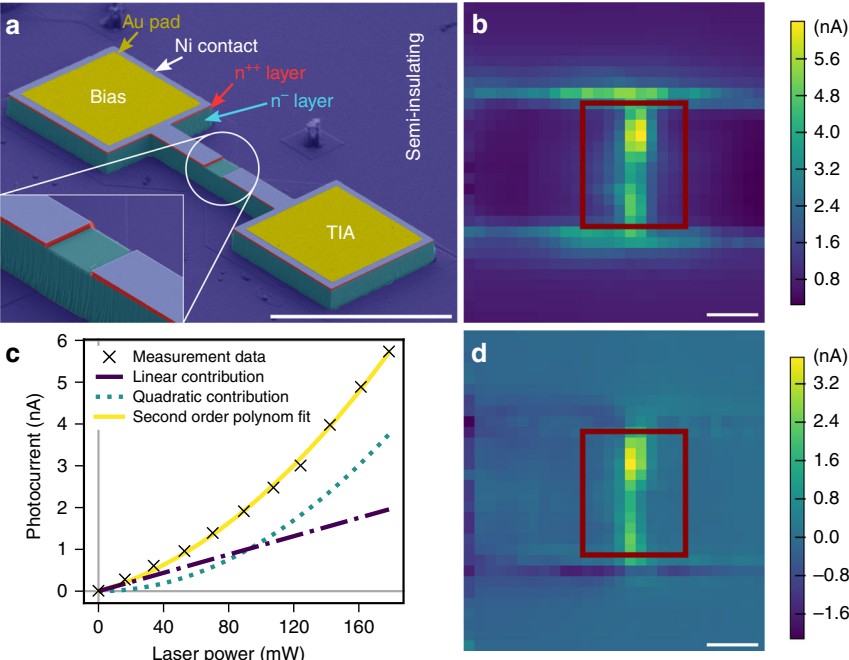

**Fig. 2** Sample and photocurrent imaging. **a** SEM picture (functional layers false color coded) of a fabricated device. Bias and transimpedance amplifier (TIA) connections are marked. Inset: Zoom-in of the etched optical opening. Scale bar: 100 μm. **b** Photocurrent map at −10 V bias. Approximate position of optical access opening marked in red. Scale bar: 5 μm. **c** Laser power dependence of the photocurrent at the position of maximum two-photon contribution corresponding to a bright spot in panel **d**. Fit $f(x) = ax^2 + bx$ separates linear and quadratic contributions, where x stands for optical power. Fit parameters are: $a = 117.77 \pm 6.43$ fA mW$^{-2}$, $b = 10.96 \pm 0.93$ pA mW$^{-1}$. Black crosses denote measurement points, yellow solid line fit $f(x)$. Purple dashed-dotted line shows linear and green dotted line quadratic contribution. **d** Map of two-photon excitation contribution to photocurrent at maximum laser power obtained via fit parameter $a$. Scale bar: 5 μm.

measurements. As the RF pulses are short (300 ns), the locking is achieved by repeating the whole spin control pulse sequences with and without RF multiple times at a lock-in frequency of 429 Hz (see Supplementary Method 1). Typical pulse lengths for optical initialization in PDMR are 600 ns laser pulse followed by 1 μs settling time.

To measure a spin-dependent photocurrent, a bias voltage is applied using a source measure unit. The resulting photocurrent is converted to a voltage by a transimpedance amplifier, which is low-pass filtered at 1 kHz. By scanning the sample position, we record photocurrent maps. At each position, we measure the photocurrent as a function of excitation power and fit the recorded data with a second order polynomial function to infer the contributions of single-photon (linear) and two-photon (quadratic) processes. For ODMR measurements, we detect fluorescence emission from 850 nm to 950 nm using a photodiode and feed the signal directly into the lock-in amplifier. All measurements are performed at a laser power of 178.5 mW (unless stated otherwise) in order to keep the same experimental conditions for PDMR and ODMR.

For PDMR measurements, the output of the transimpedance amplifier is connected to the lock-in amplifier instead of the photodiode. To avoid artefacts due to frequency-dependent coupling into the SiC device, we keep the RF frequency constant and stepwise change the magnetic field $B_0$ revealing the magnetic resonance induced signals. The magnetic field is roughly aligned along the $c$-axis of the sample. In order to map the PDMR signal, we repeatedly measure and average the PDMR amplitude. This is done by subtracting the off-resonant signal from the on-resonance data. The off-resonant signal is obtained at a $B_0$ field strength corresponding to 23 MHz detuning.

A similar approach is used for spin Rabi oscillation measurements. Here, a fixed $B_0$ field is applied and a RF field ($B_1$ field) at the spin resonance frequency drives the system, while the RF pulse length is altered and the overall sequence duration is kept constant. To account for potential RF pick-up by the lock-in scheme, we subtract an off-resonant baseline signal as described for the PDMR mapping.

**Photocurrent mapping**. Figure 2b shows the photocurrent map of our device. We find that the response is localized inside the center of the device. The spatial map of the contribution of two-photon process in the photocurrent extracted from quadratic fitting of the laser power dependency of photocurrent data (see Fig. 2c) is shown in Fig. 2d. Comparing Fig. 2b with Fig. 2d, we find that most areas show mainly linear response, indicating single photon absorption from shallow traps. In the center of the device, we observe a pronounced quadratic dependence (see also Supplementary Note 2). We perform all further measurements in this area.

**ODMR and PDMR signal analysis**. We subsequently perform stepwise $B_0$ field dependent measurements at fixed RF frequencies of 98.75 MHz and 238.75 MHz to resolve the resonances of $-1/2 \leftrightarrow -3/2$ and $+1/2 \leftrightarrow +3/2$ transitions, respectively. Both PDMR and ODMR results are shown in Fig. 3a, exhibiting the expected magnetic resonance for both ODMR and PDMR except for a small difference in the resonance positions.

We tend to attribute the shift between ODMR and PDMR resonance to offset fields present in the device, probably due to the proximity to the ferromagnetic Ni contacts (see Supplementary Note 6). As an ensemble is used and field inhomogeneities are present, lines are expected to be of Gaussian shape envelope. The linewidths in Fig. 3a are much broader in the electrical case compared to the optically detected lines. We attribute this to a

mismatch in the detection volumes for both techniques in combination with the residual magnetization of the Ni contacts. As the fluorescence light is not spatially filtered as known from confocal microscopy, the position detected by the photodiode may slightly differ from the excitation volume defining the spatial position of the electrical readout. Nonetheless, data recorded at a different position shown in Fig. 3b suggests similar linewidths for PDMR and ODMR and thus proves that broadening in Fig. 3a is not due to the PDMR technique (see Supplementary Note 3). To check that the measured signal originates from V2 centers, we measure the ground state ZFS via Zeeman splitting measurements by observing the resonances at various magnetic fields. As depicted in Fig. 3c, the ZFS is found by fitting the model function $f_{res}(B_0) = \left| ZFS \pm g\mu_B(B_0 + B_{offset}) \right|$ to the magnetic field dependence of the resonances. Thereby, $g$ is the effective electron Landé factor for $V_{Si}^-$, $\mu_B$ is the Bohr magneton and $B_{offset}$ is a magnetic field accounting for the device internal fields. With this, we find $ZFS_{ODMR} = 69.0 \pm 0.3$ MHz and $ZFS_{PDMR} = 69.1 \pm 0.3$ MHz for the optically and electrically detected case, respectively. The $g$ factors found by the fit are $g_{ODMR} = 2.02 \pm 0.01$ for ODMR and $g_{PDMR} = 2.03 \pm 0.01$ for PDMR, while the offset field is $B_{offset,ODMR} = 0.4 \pm 0.1$ G and $B_{offset,PDMR} = 3.0 \pm 0.1$ G, respectively. The presented data corroborate that the signal originates from V2 centers. As shown in Fig. 3d, the PDMR signal is located in the same area where the two-photon photocurrent contribution was found in Fig. 2d.

However, we find the sign of the PDMR signal to be dependent on the location within the device as can be seen in Fig. 3d. We tentatively attribute this to a change in the Fermi level in the device caused by charge state and ionization processes of surrounding defects[53]. Changes in the available surrounding traps can lead to different charge transport mechanisms favoring one charge species over another. Also note that two-terminal devices can exhibit complex band-bending, especially under illumination[55]. This might hinder efficient charge separation and extraction in a large portion of the device.

As a result, we cannot clearly determine if excitation from the ES or MS is responsible for the observed PDMR effect in the present device.

**Coherent control**. Next, we demonstrate coherent control, which is at the heart of advanced quantum control protocols. To this end, we first initialize the GS spin population into the ±1/2 subspace via optical excitation. Subsequently a RF driving pulse of variable length is applied to the $+1/2 \leftrightarrow +3/2$ transition. Finally the spin state is read out either optically or electrically using the next laser pulse. The latter at the same time ensures that the system is re-initialized for the following cycle. Experimental results for both ODMR and PDMR recorded under identical measurement conditions are shown in Fig. 4a. We observe Rabi oscillations with essentially identical oscillation frequency and same-order decay times from both detection methods, which indicates that PDMR has no major detrimental effect on dephasing of the continuously driven system. We further record the Rabi oscillation frequency as a function of RF field strength and observe the expected linear increase (see Fig. 4b). This proves that the PDMR of the $V_{Si}^-$ spin state in SiC allows for coherent spin manipulation and readout of the ground state and thus fulfills the fundamental requirements for more complex quantum control schemes. This is further corroborated by Hahn echo measurements (see Supplementary Note 5).

**Discussion**
To evaluate the performance of the PDMR technique, we performed a parameter dependency study (see Supplementary

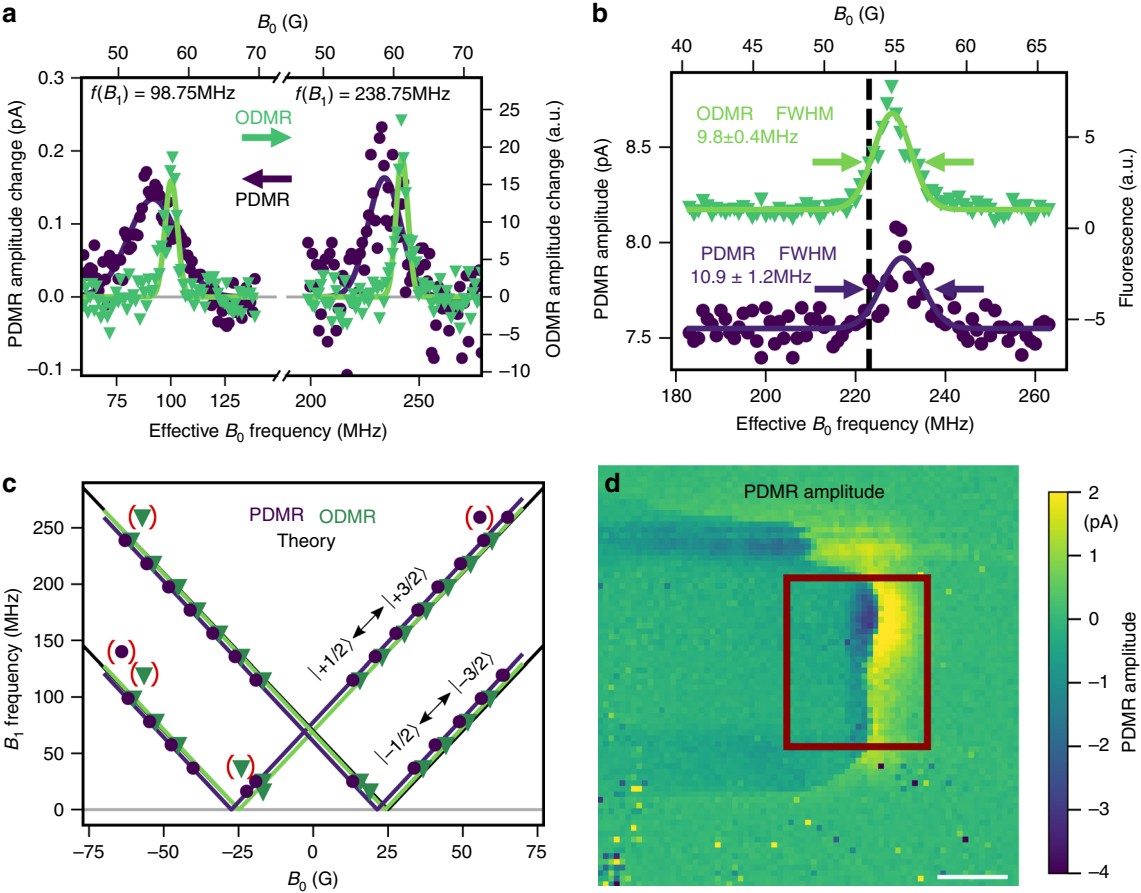

**Fig. 3** ODMR compared to PDMR. **a** Comparison of $V_{Si}^-$ PDMR and ODMR signals of upper and lower transition between $\pm 1/2$ and $\pm 3/2$ spin subsets at $-10\,\text{V}$ Bias. Offset removed for comparison. Green triangles indicate ODMR, purple dots PDMR data points. Gaussian fits to PDMR are plotted as purple, ODMR as green lines, respectively. **b** Similar linewidths in ODMR and PDMR measurements at $+20\,\text{V}$ Bias. The black dashed line indicates the expected resonance frequency. **c** PDMR and ODMR Zeeman splitting. Green triangles and purple dots mark peak positions from ODMR and PDMR measurements, respectively. Colored lines show a fit to the data, where data points in brackets are neglected. Black lines show expected theoretical values and are overlapping with measurement results. **d** PDMR amplitude map over optical excitation position. Estimated device position marked as rectangle. Scale bar: 5 μm.

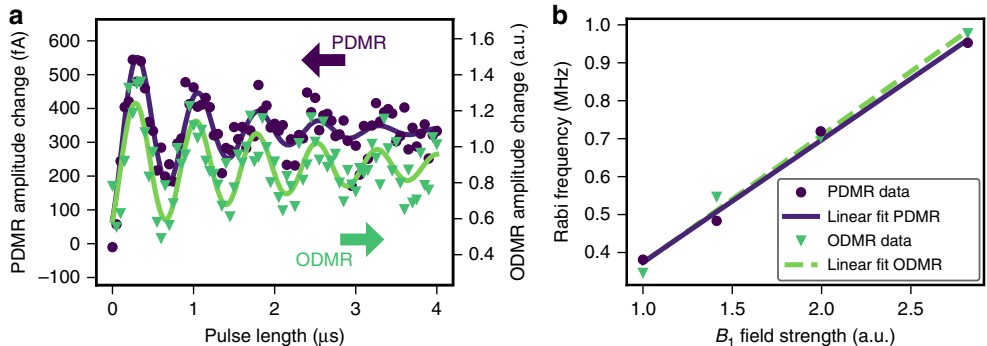

**Fig. 4** Coherent measurements. **a** Electrically detected Rabi oscillation (purple dots) directly compared to optically detected Rabi oscillation (green triangles). Purple and green solid lines are exponentially decaying sine fits for PDMR and ODMR, respectively. **b** Driving field strength dependence of Rabi oscillation frequency for optical (green triangles) and electrical readout (purple dots) with corresponding linear fits.

Notes 3 and 4). We find a ten-fold increase in SNR in ODMR compared to PDMR after normalizing to the same measurement time. In addition, the PDMR contrast is around one order of magnitude smaller than the ODMR contrast with the current device. While PDMR amplitudes are in the range of picoamperes, the mean dc background current measured by an oscilloscope parallel to the lock-in amplifier is on the order of a few

nanoamperes. This results in a typical contrast of 0.03%. On the other hand, ODMR measurements yield a contrast of around 0.1%. The background current mainly consists of the resistive current through the device due to the bias. The laser induced photocurrent also contributes to the background, but due to the pulsed type of measurement is decreased by the duty cycle. However, our measurements suggest, that we are limited by the

current experimental conditions and that multiple parameters can still be optimized (see Supplementary Note 3). Especially with increasing laser power the ODMR contrast saturates, whereas no saturation behavior is observed for PDMR yet. This is consistent with findings for NV ensembles in diamond[23]. As ionization rates are unknown and no detailed knowledge is available on the neutral charge state of the silicon vacancy ($V_{Si}^0$), future investigation of the full set of rates can help in optimization. Furthermore, refining the measurement technique and device structure can potentially improve SNR. Problems introduced by the Ni contact material can be overcome by switching to a non-standard contact material, e.g., graphene[56]. A large contribution to the noise floor is stray RF fields. We anticipate a gain in SNR by improving the device structure to be more resilient against parasitic RF coupling. In addition, the stepwise measurement was done in a conservative way and seconds of settling time between magnetic field steps were chosen in order to reach a quasi-static situation, while lock-in integration time was set to 30 ms. Using a real magnetic-field sweep or frequency-modulated RF field will speed up signal accumulation. However, due to the RF-frequency-dependent stray currents and no possibility to directly sweep the magnetic field in our experimental conditions, we have not incorporated such techniques yet. Moreover, changes to the doping profile may allow to enhance carrier extraction efficiency, but may come with the cost of an increase in background photocurrent. As the large bandgap hinders a two-photon band-to-band excitation with a 785 nm laser, the background photocurrent is likely generated by excitation of other intra-band defects created besides the $V_{Si}^-$ ensemble during the electron irradiation. In particular, this process also creates carbon vacancies ($V_C$). As $V_{Si}^-$ is verified to exist in a single negatively charged state by ODMR, we expect $V_C$ to be present in a neutral, single or double positively charged configuration, which maintains the Fermi level between $V_{Si}^-$ and $V_{Si}^{2-}$ charge states. The $V_C$ charge states can exhibit photo-ionization beginning at wavelengths around $\approx$850 nm[53,57].

As the background limits transimpedance gain, a trade-off between signal extraction efficiency and background has to be found.

Further improvements can be made by optimizing the device geometry. A smaller device should reduce capacitive coupling of RF, as well as leakage currents. This should allow for applying a higher bias voltage and amplification gain and thus better electron collection efficiency. The detection volume is limited by the maximum available laser power and presumably the band-bending inside the device. The linear absorption by other defects not only creates a background current but also reduces the number of photons available for the two-photon absorption by $V_{Si}^-$. This reduces the effective detection volume at constant laser power. Increasing the laser power and reducing the defect density in the sample should allow to enlarge the detection volume and reduce leakage currents further. Interestingly, only a small area of $\approx$2 μm × 2 μm within the aperture shows contribution to PDMR (see Fig. 3d). The contributing volume (see Supplementary Note 7 for estimation) might be enlarged by adapting a better device geometry and doping, which requires future analysis of contributions from unwanted defects and band-bending.

We find that while ODMR suffers from low photon count and collection inefficiency, PDMR can be inhibited by electron capture efficiency and noise issues. However, PDMR in 4H-SiC allows for many approaches to improve SNR in the future. To do so, further investigation of the involved processes is necessary.

To this end, we suggest to measure the dependence of the signal on excitation laser wavelength and pulse length, which might give insight into the ionization process and may ultimately improve readout fidelity and state preparation[58,59]. Since we have shown that coherent spin control of $V_{Si}^-$ can be combined with PDMR, phase interferometry type sensing protocols can be utilized, which can boost sensitivity in metrology applications by many orders of magnitude[60,61].

In summary, we have demonstrated photo-electrical readout of a $V_{Si}^-$ spin ensemble in a 4H-SiC metal-semiconductor-metal device under ambient conditions. We also report electrically detected spin coherence of this ensemble. This underlines the great potential of SiC and PDMR for quantum applications. The availability of large wafer production and processing techniques are very promising to future integration of electrical quantum devices at an industrially relevant scale. Advanced fabrication techniques can be used to integrate e.g., high-performance CMOS transimpedance amplifiers on-chip[62]. This would allow miniaturization and quantum device integration into a classical circuit design. Even integration of the optical light source might be feasible in the future[63]. Altogether, this work provides a first step towards integrated electrical quantum devices in 4H-SiC for quantum technology.

## Methods

**Photocurrent and spatial mapping**. Photocurrent is measured by applying a bias voltage (Keithley, 2636B) and exciting with a 785 nm laser (Toptica, iBeam smart), which is focused onto the device by an objective (Zeiss, Plan-Achromat ×40, NA 0.65) and measurements are performed using a lock-in type measurement (Stanford Research, SR830) after transimpedance amplification (Femto, DLPCA-200, gain of $10^8$ for PDMR, $10^9$ for Rabi). Mapping of signals is achieved by scanning the objective across the sample with a piezo-stage (Physik Instrumente, P-561.3CD).

**Optically detected magnetic resonance**. Optically detected magnetic resonance is performed by applying a fixed RF signal (Rohde and Schwarz, SMIQ03B, rubidium-referenced with EFRATOM, LPRO-101), which is pulsed (Mini Circuits, ZASWA-2-50DR) for lock-in detection and amplified (Mini Circuits, ZHL4240W) for spin control. The laser is introduced via a dichroic beamsplitter (Chroma, T810LPXR), filtered by an 850 nm optical long-pass and detected by a photodiode (Newport, Model 2151). The pulse sequence consisting of optical excitation and readout pulse, delay (1 μs) and RF driving pulse is repeated multiple times. Subsequently, the same pulse train is repeated with the RF turned off. The lock-in is locked to the frequency at which the pulse sequences are applied with and without RF. Thus, the signal detected by the lock-in amplifier is a RF induced fluorescence. Magnetic field is stepwise increased over the expected spin resonance frequency, taking fluorescence light measurements at each magnetic field.

**Photocurrent detected magnetic resonance**. Photo-electrical detected magnetic resonance is performed similar to the optically detected case. Instead of reading the photodiode voltage, the lock-in input is connected to the transimpedance amplifier output. Now, the detected signal is the change in photocurrent induced by the RF.

**Detection of Rabi oscillations by ODMR and PDMR**. The coherent control is performed by varying the RF pulse length. The total control pulse length is fixed by adding a waiting time between the RF pulse and the start of the next pulse train for shorter RF pulses. For this, the pulse sequence consisting of initialization and spin control pulse is repeated multiple times with RF switched on and then repeated with RF switched off. Thus the signal is locked to RF influence of fluorescence or photocurrent for ODMR and PDMR, respectively. Measurements are performed at an on-resonance magnetic field and RF frequency condition, as well as at an off-resonance by detuning the magnetic field. The obtained data is then substracted from each other to remove background.

## Data availability
The data sets generated during and/or analyzed during this study are available from the corresponding author on request.

## Code availability
All code used for analyzing the raw data is available upon request from the corresponding author.

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

## Acknowledgements

The authors thank Florian Kaiser for his valuable help in preparing the paper. We acknowledge financial support by the EU (ASTERIQS and ERC SMeL), BMBF (BrainQSens), the Max Planck Society, the Volkswagen Foundation, the Swedish

Research Council (VR 2016-04068), the Carl Tryggers Stiftelse för Vetenskaplig Forskning (CTS 15:339), the Swedish Energy Agency (43611-1), the Knut and Alice Wallenberg Foundation (KAW 2018.0071), the Korea Institute of Science and Technology institutional programs (2E27231, 2E29580) and the Japan Society for the Promotion of Science KAKENHI (17H01056, 18H03770).

## Author contributions

The initial planning of the project was done by M.N., M.W., S.-Y.L, N.T.S., and J.W. J.U.-H., and N.T.S. designed the device structure and performed the sample growth. R.S. fabricated the device. S.O., T.O., and J.I. planned and performed electron irradiation. M.N. designed and performed all experiments. M.N., T.R., M.W., A.M., N.M., Y.-C.C., S.-Y.L., and J.W. analyzed the data. The paper was written by M.N., M.W., T.R., R.S., and J.W. with contribution from all of the authors.

## Competing interests

The authors declare no competing interests.
