## [Transparent Peer Review File · Nature Communications]

Reviewers' comments:

Reviewer #1 (Remarks to the Author):

This paper demonstrates the possibility of using photocurrent to read out the ensemble spin of silicon vacancy centers in 4H-SiC at room temperature. By comparing this novel technique (PDMR) with the well-known ODMR technique along with a demonstration of electrically detected spin coherence, authors claim the qualification of SiC and PDMR for quantum applications, which is further substantiated by the maturity of SiC industry and the associated practical applications [Phys. Rev. Applied 8 044015 (2017)]. There is no doubt that electric readout would facilitate miniaturization of the device. However, in the respect of the performance, PDMR seems not be able to overtake ODMR, and its advantages are not clearly conveyed in this paper.

One of main motivations cited for PDMR is its capability of circumventing the low photon collection efficiency that is encountered by ODMR. Instead, PDMR is limited by another physical quantity, i.e., electron (or hole) collection and detection efficiency, which can be significantly compromised due to the presence of various defects in the sample, band distortion at the electrode contacts, and finite electron detection efficiency. As pointed out by the paper, the SNR of PDMR is 10 times lower than the SNR of ODMR, justifying the opposite point of view that ODMR is more competent and reliable for spin readout in 4H-SiC system. In addition, many quantum applications utilize single confined spin rather than an ensemble of spins as fundamental qubit to establish quantum architecture, since single spin possesses a longer coherence time comparing to the latter thanks to the elimination of environment-induced inhomogeneous dephasing that is almost always present in the solid-state qubit. So far, single-shot readout has been realized on single confined spin by using ODMR technique. Is it possible read out a single electron spin with PDMR? How was it compared to ODMR on single spin readout?

If authors agree to make modifications on motivation, improve the comparison with ODMR, and address the following concerns, I recommend the paper to be published in Nature Communication.

Many quantum applications (such as metrology, computation and communication) impose high-speed operation on qubit and quantum memories. For this concern, what is the bandwidth of PDMR method? Is it limited by the electron extraction process or the process for reestablishing the charge balance? How is it compared to ODMR? As mentioned in Page 4, there are three different mechanisms for reestablishing the steady-state charge distribution, i.e., electron recapture from CB, from nearby defects, or from VB. Since only the last mechanism is shown in Fig. 1c, does it imply that the process of charge redistribution is dominated by the last mechanism? Please comment on it.

Is there any scalable fabrication of such silicon vacancy towards their applications?

What is branching ratio of exciting an electron into CB as compared to ISC relaxation? On page 4, authors make a statement that the spin-dependent contribution to photocurrent is expected to be the sum of currents created by promoting an electron either from the ES or MS to the CB.

However, if the electron can be promoted from ES and MS to CB, there would be less contrast or even no difference in photocurrent when exciting populations in $|\pm 3/2\rangle$ or $|\pm 1/2\rangle$ manifolds. Please clarify your point.

Is there any thumb of rule for choosing the size of active area? It is not clear that if the different look of photocurrent map in Fig. 2(b) comparing to fluorescence map in Fig. S3 is caused by using a spatial filter. If so, what is the function of the spatial filter? And where is it placed in experiment? If not, why the two maps look so different?

What is the physical process underneath the single-photon induced (linear) photocurrent? In Fig. 2(d), the coefficients of two-photon excitation are negative values for some areas of mapping.

Does it correspond to a reverse of direction of two-photon induced current? How would it happen?

Regarding the Rabi oscillations as shown in Fig. 4(a), is it possible to extract the dephasing rates from the data? It seems to me that PDMR decays at a faster rate than ODMR does.

Minor modifications: please swap the labeling sequence of Fig. 3(b) and Fig.3(c) for a better reading experience by following the appearance of the figures in the text. Besides, please use a different color for black dotted lines in Fig. 3(b) for clarity.

Reviewer #2 (Remarks to the Author):

The manuscript NCOMMS-19-11805, "Coherent electrical readout of defect spins in silicon carbide by photo-ionization at ambient conditions" by Niethammer et al. reports on the demonstration of both optically detected magnetic resonance (ODMR) as well as photoconductivity detected magnetic resonance (PDMR) of silicon vacancy center (VSi-) spin states in electronic devices made out of 4H-SiC metal-semiconductor-metal hetero structures at ambient conditions. In particular, the manuscript is focused on the demonstration that the magnetic resonant spectra of two low magnetic field/frequency spin transitions within the $s=3/2$ point defect system (occurring at approx. 100MHz and 240MHz, respectively) can be measured, indicating, that electric current in this material can be governed by spin-dependent recombination and, thus, that electric current can be used to probe the spin state of paramagnetic point defects. In essence, the manuscript reports on the experimental demonstration of an electronic process in 4H-SiC which can be used to implement an electrical VSi- spin readout, and that this readout can be implemented using a sequence of optical charge carrier injection pulses combined with radio frequency (RF) magnetic resonance probe and a lock-in detection scheme.

The manuscript also reports on coherent control experiments demonstrating electrical detection of spin-Rabi oscillation and, thus, of electrical detection of coherent spin motion. The latter is a crucial prerequisite for the utilization of the studied spin-dependent electronic transition for quantum information applications.

Overall, I find the results presented in this manuscript valid, of high scientific quality and of great interest for readers from a broad range of backgrounds. Thus, I deem the manuscript, of great significance and eventually publishable in Nature Communications. After reading the manuscript and then, reading it again, I have had a few questions about the results presented in the manuscript and also about the presented experiments that led to these results. A few aspects of this study have either not been addressed or not been properly addressed. I am listing these in the following, together with some minor comments about display items:

- 1) Overall, I find the discussion of the spatially resolved photoexcitation as well as the ODMR detection scheme (the optical setup, technical aspects, etc.) insufficient. A bit more detailed description would help here a lot, as the reader's understanding about the interpretability of the data depend on this. Possibly, a sketch would be great. When I tried to get more information about this after reading the main text, I looked at the SI and ended up being quite confused by Figure S4: What was the location difference between the Focus planes for the data in the left (orange framed) column versus the right (red framed) column? What exactly is plotted in the center columns of the figure, the so called z-slice? I understand that the data at the orange and red marked slices somehow pertain to the data on in the left and the right columns, but how exactly are they related? What are scales on which the focal point was shifted along the z-axis?
- 2) I miss a bit a discussion about heterogeneity of the observed PEDMR signals as a function of position within the sample.
- 3) One issue that I find not discussed conclusively is the question of whether the observed spin-dependent recombination signals are truly due to the VSi- centers or due to other defect species. The obtained current and light detected magnetic resonance spectra are not suitable to produce accurate g-factors or g-tensors due to the employed low magnetic fields, the low signal to noise ratio as well as the significant random magnetic field distributions throughout the sample caused by the Ni layer. I understand that the argument for this defect is the quadratic dependence of the photocurrent from the Laser power but can we exclude that other defects show a similar two-transition behavior as the VSi-. If so, then I'd recommend being a bit more explicit and really summarize what makes the observed signals being unambiguously attributable to the VSi- center. Also, what was the reasoning behind the employed RF frequencies. I would think that by using much higher frequencies, a much better spectroscopic analysis of the measured magnetic resonance spectra would be possible.
- 4) I understand that the Ni layer in the used samples is needed to produce a Schottky contact and

also, as explained in the SI, it is needed as etch stop during the sample preparation. I wonder though whether there could be other, non-ferromagnetic materials that could work in a similar way? If not, I would like the manuscript to be more explicit in the main text about why the sample has to be designed that way it is..

5) When it comes to figures, I would shift panel (d) in Fig. 1 to into Fig. 3 as the experiment and the chosen readout sequence are not really discussed before.

6) In figure 3, I would not use dashed and dotted lines. The fit plots are hardly distinguishable from the data points.

7) I would have appreciated a discussion about the observed coherence times of the resonantly excited and electrically read spin states. Rabi oscillation, of course, does not provide exact information about T2 spin relaxation times. However, it does allow for an estimate and a discussion of T2, i.e. a discussion of what interaction of the paramagnetic defect could potentially be coherence time limiting. For instance, from the measurement of the random magnetic field distribution as shown in Fig. S8 of the SI, one may or may not be able to determine whether the observed Rabi oscillation dephasing is due to coherent dephasing throughout the ensemble of VSi-centers or whether it actual is due to real coherence decay. If the former turned out to be the case, the electrically detectable spin coherence times could even be longer than the microsecond scale which they appear to have and that would make the outcome of this study even more significant.

Reviewer #3 (Remarks to the Author):

The authors report the photo-electric detection of the electron spins associated with silicon vacancy ensembles in SiC. They demonstrate photocurrent detected magnetic resonance (PDMR) of coherently controlled spins and compare their findings to the more conventional optically detected magnetic resonance (ODMR) spin readout. The manuscript is reasonably well written and organized, and the data seems to be of good quality, although I do have a few comments regarding the details which may help clarify the presentation.

The main message in this manuscript appears to be that PDMR offers a scalable alternative to ODMR techniques. While these techniques have been demonstrated in systems like the NV center in diamond, the maturity of SiC electronics offers great promise in scaling the materials and devices for quantum applications. The authors present compelling data for the PDMR detection but offer little in the way of understanding regarding the physical underlying mechanism.

Specifically –

i) The discussion of the mechanism behind the current increase and decrease (page 4-5 of main text) is poorly explained. A discussion beyond just simply acknowledging the competing rates would give better intuition into the technique.

ii) The discussion regarding the spatial dependence of the PDMR signal and the relative position of the optical spot (Figure 3d) is not well explained. What are the possible underlying mechanism behind this PDMR spatial dependence?

iii) Please provide better intuition regarding the surprising change in sign of the PDMR signal (page 11). What are the possible underlying physical mechanisms that contribute to this sign change? What surrounding defects might affect this, and how might these effects be mitigated?

After successfully addressing a few of these comments, publication in Nature Communications will be recommended from my side.

I also have a number of minor issues/comments which are intended to make the presentation clearer and easier to follow:

- a) Related to point i) above, in the main paragraph on page 4 starting with 'In the following, ...', there is a lot of detailed description about the underlying charge dynamics but the discussion is worded and formatted in a confusing fashion. While the paragraph does refer to figures 1(a) – (c) as a whole, a proper breakdown and description of each of the different possible photo-ionization mechanisms will bring clarity.
- b) The discussion regarding the ISC rates (page 4) could use a more detailed discussion, possibly in the SI.
- c) When referring to the supporting information, you could also include specific sections for clarity and convenience.
- d) How does the reduced contact quality help minimize the dark current? (Page 6)
- e) What is the motivation behind the dimensions of the $n^{++}/n^{-}n^{++}$ structure? (Page 6) Why did you use this type of doped structure?
- f) Is there a specific reason why 429 Hz was chosen as a lock-in frequency? (Page 8)
- g) The phrase 'The beam in front of the photodiode is attenuated by an iris to prevent detector saturation' (page 8) is unclear and out of place. Firstly, this feels like a detail that could be described in better in the supplemental, and further motivated as to the use of an iris (as opposed to a neutral density filter) and what sort of attenuation is expected and needed.
- h) You mention 'the magnetic field is roughly aligned along the c-axis' (Page 8). How susceptible is this effect to field misalignment in general? Do you attribute any PDMR effect to changes in the B-field alignment during the sweep?
- j) My understanding is the off-resonant signal is obtained as a single magnetic field (corresponding to 23 MHz). Why was this not done at all steps of B-fields?
- k) What is the estimated detection volume? (page 9) You reference the sample geometry but is the volume assumed to be the wide of the gap and the thickness of the n- or a subset thereof?
- l) Regarding the phrase 'likely generated by excitation of other intra-band defects ...' (Page 13) - Are there any indications what those other defects might be? Can specific defects be attributed to the spatial dependence or the sign change observed in the PDMR signal?
- m) The discussion regarding the detection volume and leakage currents (page 13) could be expanded. What currently limits the detection volume? What is the estimate 'small area within the aperture' that shows PDMR contribution? I understand this might be a complex process but is there a possible intuitive picture that might be discussed or is this not well understood?

Supplemental:

- n) Please add section markings to each section and refer specific discussions in the main text to the requisite section. This makes things far easier to follow
- o) What is the intent of the post-irradiation I-V characteristic plot? (Figure S2b) Is the intent to show the effect of irradiation-induced defect? This would be a good place to discuss what other possible defects might contribute.
- p) In the SI (S-5) you detail a focal dependence of the PDMR signal inside the device. Expanding on this sort of discussion would be useful in addressing general comment (ii) as well as the comment on the estimated detection volume (k) and (m). How does the small excitation volume relate to the detection volume? What would you estimate the minimum illuminated defect density you would need to find PDMR signal?
- q) The discussion related to the SNR ratio is confusing (Page S-9). Specifically, the discussion related to the lock-in constant offset and the various definitions of 'contrast' is not clearly detailed.

We thank all Reviewers for their valuable comments and remarks.

In the following we present point-to-point answers to all the questions raised by the reviewers and describe by which means we translated their remarks and comments into an improved manuscript.

Answers to Reviewer #1:

Comment 1-a: This paper demonstrates the possibility of using photocurrent to read out the ensemble spin of silicon vacancy centers in 4H-SiC at room temperature. By comparing this novel technique (PDMR) with the well-known ODMR technique along with a demonstration of electrically detected spin coherence, authors claim the qualification of SiC and PDMR for quantum applications, which is further substantiated by the maturity of SiC industry and the associated practical applications [Phys. Rev. Applied 8 044015 (2017)]. There is no doubt that electric readout would facilitate miniaturization of the device.

Answer: We appreciate the reviewer's accurate understanding for the motivation and significance of the studied electrical readout methods of defect spins in semiconductors. We added the stated reference to the list of applications in the introduction (changelog M7).

Comment 1-b: However, in the respect of the performance, PDMR seems not be able to overtake ODMR, and its advantages are not clearly conveyed in this paper.

One of main motivations cited for PDMR is its capability of circumventing the low photon collection efficiency that is encountered by ODMR. Instead, PDMR is limited by another physical quantity, i.e, electron (or hole) collection and detection efficiency, which can be significantly compromised due to the presence of various defects in the sample, band distortion at the electrode contacts, and finite electron detection efficiency. As pointed out by the paper, the SNR of PDMR is 10 times lower than the SNR of ODMR, justifying the opposite point of view that ODMR is more competent and reliable for spin readout in 4H-SiC system.

Answer: We thank the reviewer for pointing out this issue and agree that many limiting factors can occur in the PDMR technique as well. Nevertheless we want to mention, that to the best of our knowledge, this is the first time PDMR has been shown defects in SiC. No optimization steps have been taken to maximize the signal from fabrication aspects yet, as first a proof-of-principle and better understanding of the functionality needs to be obtained. We attribute the low SNR of PDMR (compared to ODMR) to BG current induced by photoionization of parasitic defects, such as carbon vacancies V_C . Avoiding unwanted V_C during [J. Cryst. Growth 281, 370 (2005)] and after growth [Appl. Phys. Express 2, 41101 (2009)], combined with localized V_{Si} creation [Nano Lett. 19 2377 (2019), ACS Photonics 6, 7, 1736-1743 (2019), Phys. Rev. Appl. 7, 64021 (2017)] may potentially lead to a better SNR, as shown for the NV center in diamond [Nat. Commun. 6, 8577 (2015), Science 363, 6428, pp. 728-731 (2019)]. Since we have shown the feasibility of the PDMR, we expect that our study will trigger the beginning of many related investigations, which will allow to find the optimized device structure and fabrication process.

Action taken: Emphasized improvement of SNR in PDMR over ODMR in NV case (changelog M21), more clearly stating motivation behind new detection (changelog M22), improved approaches for better SNR and added limitations of ODMR and PDMR (changelog M23).

Comment 1-c: In addition, many quantum applications utilize single confined spin rather than an ensemble of spins as fundamental qubit to establish quantum architecture, since single spin possesses a longer coherence time comparing to the latter thanks to the elimination of environment-induced inhomogeneous dephasing that is almost always present in the solid-state qubit.

Answer: We do agree with the advantage of utilizing isolated defect spins. As this is the first proof-of-principle demonstration of PDMR in 4H-SiC, we used an ensemble, which has benefits in signal strength. We want to mention, that also for NV in diamond, PDMR has first been shown in an ensemble approach [Nat. Commun. 6, 8577 (2015), Phys. Rev. Lett. 118, 037601 (2017), Phys. Rev. B 95, 041402(R) (2017), Phys. Rev. Applied 7, 044032 (2017)]. Recently, single defect sensitivity has been demonstrated [Science 363, 6428, pp. 728-731 (2019)]. To reach single defect sensitivity as a long-term goal, we first want to investigate fundamental mechanisms, such as the spin-dependent ionization process and impacts by other defect, which will allow us to find optimal PDMR protocols for this material. In addition, we plan to find an optimal device structure such as a p-i-n junction device, in which the silicon vacancies can be specifically created only in the pure i-layer by e.g. laser defect writing [Nano Lett. 19, 2377 (2019)]. Then, we will definitely aim to reach the single spin level.

Comment 1-d: So far, single-shot readout has been realized on single confined spin by using ODMR technique. Is it possible read out a single electron spin with PDMR? How was it compared to ODMR on single spin readout?

Answer: Although this is an interesting topic, it is far beyond the scope of our current work. Thus we don't provide the related discussion in the revised manuscript. However, since this is a very interesting question, we want to make some short comments on this topic:

There are several approaches to realize single shot readout for the NV center in diamond: Single-shot readout of the electron spin is possible for the NV center in diamond at cryogenic temperatures, where single spin-selective and -preserving transitions can be addressed optically [Phys. Rev. Lett. 114, 136402 (2015), Nature 477, 574–578 (2011)]. In this sense the readout is not accomplished by common known ODMR as it is used for most room temperature NV experiments, as ODMR readout is not spin state preserving. In the same analogy also the PDMR technique is not suitable for this type of experiment, as there is no spin state preservation during recharging. However, the NV center electron spin allows for single-shot readout of nuclear ancilla spins [Science 329 542 (2010), Nature 506 204 (2014)]. As this is accomplished by conventional ODMR readout, this should also be feasible by utilizing PDMR. Especially, it has been demonstrated that the spin state of an axillary nuclei is preserved even while changing the defects charge state [Nano Letters 17 5931 (2017)]. However, this relies on very good SNR. At the moment, we think that the state-of-the-art PDMR on NV in diamond should allow for such measurements already, while the SNR in 4H-SiC in both ODMR and PDMR must be improved further to address hyperfine transitions efficiently.

Comment 1-e: If authors agree to make modifications on motivation, improve the comparison with ODMR, and address the following concerns, I recommend the paper to be published in Nature Communication.

Answer: As advised, we provide improved writings for the motivation, and more discussions about the limiting factors for the efficiency of the presented PDMR protocols and how to circumvent them as above and following.

Action taken: Clarified motivation, improved explanation of approaches to increase SNR and added conclusion about ODMR and PDMR limitations in SiC (changelog M21-M23).

Comment 1-f: Many quantum applications (such as metrology, computation and communication) impose high-speed operation on qubit and quantum memories. For this concern, what is the bandwidth of PDMR method? Is it limited by the electron extraction process or the process for reestablishing the charge balance? How is it compared to ODMR?

Answer: In the ODMR case, establishing a room-temperature spin polarization by optical pumping takes at least one, better multiple ISC cycles. This adds up to a few hundred nanoseconds. In ODMR, the readout is established within another several hundred ns of the next laser pulse until the next steady-state spin polarization is achieved. Together with typical pulse lengths for spin manipulation, the minimum pulse duration adds up to about 1 μ s leading to a best case bandwidth of roughly 500 kHz - 1 MHz.

For the PDMR, the ISC cycle also has to be repeated at least once for spin polarization to build up. However, we assume charge dynamics to be much faster than cycling. Indeed we used the same experimental timings in ODMR and ODMR. Thus, we deem that the bandwidth should be of the same timescale and the duration of spin operations is not affected.

Action taken: Added estimation of achievable bandwidth in section S16 [changelog S3].

Comment 1-g: As mentioned in Page 4, there are three different mechanisms for reestablishing the steady-state charge distribution, i.e., electron recapture from CB, from nearby defects, or from VB. Since only the last mechanism is shown in Fig. 1c, does it imply that the process of charge redistribution is dominated by the last mechanism? Please comment on it.

Answer: An electron recapture from the conduction band should not lead to a net current, thus will not be detected by our measurement scheme. Recapture from other defects cannot be ruled out, as especially V_C defects are present in the sample and known to be good charge traps [Appl. Phys. Lett. 88, 052110 (2006)].

We think it is not trivial to experimentally obtain recharging rates at the current state of technology. Additionally, we think that evaluating whether the V_{Si} is directly recharged from the valence band, or another defect is the source of recharge will require a detailed analysis including device fabrication variation.

In order to not further complicate Figure 1(c) but provide a reasonable recharging mechanism, we did only incorporate the direct valence band optical recharging.

In a first approximation, we treat the recharge of V_{Si} from the VB as a photoionization of an electron from the VB to the V_{Si} . Thus, this process can be understood as a fast process. In contrast, the electron recapture from nearby defects may be either slow or fast process depending on the charge state of the trap defects. In this work, the most possible defects are V_C defects and they are likely in the singly positively charged state. Ionization of V_C^0 provides electrons. However, the recapture of the electrons by V_{Si}^0 centres will be a slow process compared to the ionization process. Therefore, we identify recharging from the VB as the most plausible dominant process [Phys Rev B 98, 195202 (2018)].

Action taken: Revised PDMR explanation (changelog M27-M29,M51-52).

Comment 1-h: Is there any scalable fabrication of such silicon vacancy towards their applications?

Answer: The fabrication of such defects has been studied in several publications in the last few years. Foremost, in our manuscript electron irradiation is used to create silicon vacancy centers. This process is readily available and can irradiate large sample sizes. Furthermore, ion beam [Phys. Rev. Appl. 7, 64021 (2017), Nano Lett. 2017,17,5,2865-2870] and neutron irradiation [Nat. Commun. 6 7578 (2015)] have been studied and the former has been used to write small clusters of defects. Additionally, we have recently demonstrated that the vacancies can also be created by high-energy pulsed laser beams [Nano Lett. 19 2377 (2019)]. By this, controlled and scalable processes for defect creation are available, although not as intensively studied as for other materials yet.

Action taken: Added a sentence about scalable fabrication methods for V_{Si} in the revised manuscript (changelog M6).

Comment 1-i: What is branching ratio of exciting an electron into CB as compared to ISC relaxation?

Answer: In order to establish a comprehensive rate model for PDMR further experiments on the excitation rates into CB are necessary, and they need to be combined with known rates [Nat. Commun. 6, 7578 (2015), Phys. Rev. Applied 11, 024013 (2019)]. We do agree on the importance of a comprehensive model for the PDMR in SiC, and we plan to systematically investigate in the future. Note, that to the best of our knowledge the (ionized) neutral charge state of V_{Si} has not been detected optically, which makes investigations difficult at this moment.

Action taken: Added a brief note mentioning this aspect of ionization for future models (changelog M19).

Comment 1-k: On page 4, authors make a statement that the spin-dependent contribution to photocurrent is expected to be the sum of currents created by promoting an electron either from the ES or MS to the CB. However, if the electron can be promoted from ES and MS to CB, there would be less contrast or even no difference in photocurrent when exciting populations in $|\pm 3/2\rangle$ or $|\pm 1/2\rangle$ manifolds. Please clarify your point.

Answer: The metastable state of the V_{Si} in 4H-SiC has not been studied extensively and their electronic energy levels and level structures are not known accurately yet. Thus, the possibility of optical excitation from the metastable state to the conduction band cannot be ruled out. We included this process in the discussion for the sake of completeness. However, as this process would lead to a sign reversal and thus reduce the photocurrent resonance peak. The sign of the of PDMR peak may provide information on which of the processes is more dominant.

Unfortunately, with the obtained data and ambiguity in sign in the observed PDMR signal, we cannot precisely conclude on the metastable contribution. Additional experiments will be helpful, especially wavelength dependent measurements at low temperature might give more information on the metastable state, e.g. energetic position within the bandgap. Nevertheless, as this property is not directly accessible by ODMR technique, PDMR might prove to be a valuable tool in this regard in the future. We improved on the explanation of our argumentation given in the manuscript.

Action taken: Revised PDMR explanation (changelog M27-M29, M51-M52).

Comment 1-m: Is there any thumb of rule for choosing the size of active area?

Answer: Ideally, the active device area should be limited by the two-photon absorption focal volume in order to maximize SNR. However, sufficient optical excitation power will be required. As the desired power densities were not known to us prior to this study, we designed devices in various sizes. We used a $10\mu\text{m} \times 12\mu\text{m}$ active area, while a single-photon confocal spot-size is approximately $1\ \mu\text{m}^3$. Because the illumination near the metal contacts might have led to additional effects, we used a larger active area that allows for some distance from the contacts and scanning of the device center to obtain spatial information as well.

Comment 1-n: It is not clear that if the different look of photocurrent map in Fig. 2(b) comparing to fluorescence map in Fig. S3 is caused by using a spatial filter. If so, what is the function of the spatial filter? And where is it placed in experiment? If not, why the two maps look so different?

Answer: The two figures show different data. Fig. 2(b) represents the measured photocurrent in which the spatial position represents the position of the excitation laser. Fig. S3 shows the fluorescence of the device using a photodetector, which ideally collects the signal from a confocal spot. As photocurrent depends also on the band-bending and defect centers, the signal looks different than the homogeneous fluorescence map. We use a photodiode and no additional spatial filter (like a pinhole) in order to obtain better ODMR SNR. Thus our configuration does not provide maximum spatial filtering as one would expect for a confocal arrangement. In consequence, a measurement at the same excitation position might be slightly shifted in optical detection in respect to electrical detection.

Action taken: Added measurement sketch for clarity over the experimental setup (changelog S2) and better explanation in main text (changelog M54).

Comment 1-o: What is the physical process underneath the single-photon induced (linear) photocurrent?

Answer: We attribute the linear current contribution from absorption of defects in SiC, such as V_C , which can be ionized via single photon absorption at the wavelength used. See answer to **Comment 3-q** for more details.

Action taken: Added paragraph on most likely defects for linear photocurrent contribution (changelog M30).

Comment 1-p: In Fig. 2(d), the coefficients of two-photon excitation are negative values for some areas of mapping. Does it correspond to a reverse of direction of two-photon induced current? How would it happen?

Answer: As can be seen in Fig. S4, the PDMR amplitude seems to correspond to an opposite sign as the two-photon current. At this point we have to admit that we do not fully understand the local change in sign. This also has been observed in PDMR on NV in diamond and attributed to change in charge states of surrounding defects due to the illumination. We hope to find more insight in the physical process in temperature and wavelength dependent measurements. Please also see answer to **Comment 2-b**.

Action taken: Added paragraph on spatial dependence of PDMR signal (changelog M24).

Comment 1-q: Regarding the Rabi oscillations as shown in Fig. 4(a), is it possible to extract the dephasing rates from the data? It seems to me that PDMR decays at a faster rate than ODMR does.

Answer: The dephasing times on the Rabi signals are $2.2 \pm 0.5 \mu\text{s}$ ODMR and $1.1 \pm 0.1 \mu\text{s}$ for the PDMR as obtained from the exponentially decaying sinusoidal fit. Although these do not match completely, the dephasing is not seriously increased. Such changes can happen due to the non-uniform magnetic field introduced by the nickel layer. Since the dephasing time can be influenced by many experimental and environmental parameters, we directly measured T_2 by Hahn echo measurements. From this we estimate a minimum coherence time of $7 \mu\text{s}$ for electrical and optical measurements at low field. At higher field we can track coherence at least for $14 \mu\text{s}$. As the PDMR readout electronics (TIA) is connected also during the ODMR measurement and is just not fed to the lock-in amplifier, we performed ODMR Hahn echo at 0 V and 20 V bias to demonstrate that the applied bias does not affect the coherence. All combined this strongly suggests that the PDMR mechanism does not lead to degradation of coherence time. The results and discussion are added to the supporting information.

Action taken: Measured T_2 times, added section “7 Hahn-Echo measurements” with discussion to the Supplementary as Discussion 5 (changelog S4,S22, M25).

Comment 1-r: Minor modifications: please swap the labeling sequence of Fig. 3(b) and Fig.3(c) for a better reading experience by following the appearance of the figures in the text. Besides, please use a different color for black dotted lines in Fig. 3(b) for clarity.

Answer: We thank the reviewer for this suggestion. We improved the styling of Fig. 3(b) and swapped the panels according to appearance in text.

Action taken: Panels (b) and (c) of Fig. 3 swapped. Improved distinguishability of old Fig. 3(b) (changelog M2-M4, M13-M16).

Answers to Reviewer #2:

Comment 2-a: [...] A few aspects of this study have either not been addressed or not been properly addressed. I am listing these in the following, together with some minor comments about display items:

1) Overall, I find the discussion of the spatially resolved photoexcitation as well as the ODMR detection scheme (the optical setup, technical aspects, etc.) insufficient. A bit more detailed description would help here a lot, as the reader's understanding about the interpretability of the data depend on this. Possibly, a sketch would be great. When I tried to get more information about this after reading the main text, I looked at the SI and ended up being quite confused by Figure S4: What was the location difference between the Focus planes for the data in the left (orange framed) column versus the right (red framed) column? What exactly is plotted in the center columns of the figure, the so called z-slice? I understand that the data at the orange and red marked slices somehow pertain to the data on in the left and the right columns, but how exactly are they related? What are scales on which the focal point was shifted along the z-axis?

Answer: We thank the reviewer for this valuable remark. Indeed, during revision of our manuscript we realized that a sketch will improve the readers experience and, hence, the presentation of the results.

As explained in section SI4, the z-slices are vertical cross sections along the z-axis, fixing the y and varying the the z coordinates. The horizontal xy-slices are taken at the focusing depth as plotted in the z-slice with the z coordinate fixed. The scale bar shows that the difference in focus is about 17 μm .

We agree that the presentation of the data can be more clear and added indicating arrows to the figure.

Action taken: Section with description and sketch of the experimental setup added as the Supporting Methods 1 (changelog S2). Improved Fig. S4 (changelog S7) and description (changelog S8,S9).

Comment 2-b: 2) I miss a bit a discussion about heterogeneity of the observed PEDMR signals as a function of position within the sample.

Answer:

Unfortunately, so far, we cannot experimentally identify the origin of this heterogeneity. We expected to see photocurrent and PDMR from the whole optically accessible area. Clearly this is not the case. Therefore we suspect the following mechanisms:

Carrier mobility/diffusion length can have an impact, which can be largely affected by the high density of V_C created by high-dose electron irradiation, which can pin the Fermi level. Thus, the band-bending in combination with traps/defects might be the most crucial parameters in this regard. Note, that the effective internal electric field due to the applied bias can behave unexpectedly, especially under illumination ([Nat. Electron 1, 502–507 (2018)]). Therefore, the efficiency of charge separation and extraction might be modulated locally thus affecting photocurrent and resulting in a heterogeneity.

The change in sign can be related to surrounding defect charge states, as the recombination and recharging processes can be different depending on the band-bending and surrounding defect charge states.

Photophysics have been studied at low-temperature at presence of divacancy defects, which should not be abundant in our device since high temperature annealing, necessary for creation of divacancies, was not performed [<https://www.nature.com/articles/s41467-017-01993-4.pdf>].

From this, a working theory is that depending on the charge capture of surrounding defects, the photoionization can provide an electron to the conduction band and then a hole in the valence band can be created. Depending on the defects around, hole recombination or electron recombination can be more likely altering the charge carrier mobility for the different charge particles. By this, charge extraction efficiency and mobility can depend on the local position in respect to band-bending, thus change the sign of the net current.

However, no details on this process are available yet. This overall drastically complicates the underlying physical process. We are open for any suggestions, how this can be measured or modeled and insight into the physics can be gained, however at the moment we feel that no model can be stated that would accurately describe the situation.

Action taken: Added paragraph on spatial dependence of PDMR signal (changelog M24).

Comment 2-c: 3) One issue that I find not discussed conclusively is the question of whether the observed spin-dependent recombination signals are truly due to the V_{Si} - centers or due to other defect species. The obtained current and light detected magnetic resonance spectra are not suitable to produce accurate g-factors or g-tensors due to the employed low magnetic fields, the low signal to noise ratio as well as the significant random magnetic field distributions throughout the sample caused by the Ni layer. I understand that the argument

for this defect is the quadratic dependence of the photocurrent from the Laser power but can we exclude that other defects show a similar two-transition behavior as the V_{Si^-} . If so, then I'd recommend being a bit more explicit and really summarize what makes the observed signals being unambiguously attributable to the V_{Si^-} center. Also, what was the reasoning behind the employed RF frequencies. I would think that by using much higher frequencies, a much better spectroscopic analysis of the measured magnetic resonance spectra would be possible.

Answer: Both the ODMR and PDMR signals are attributed to the V_{Si^-} centers, as to the best of our knowledge no other defects have been reported to have a zero-field splitting (ZFS) of 70 MHz in 4H-SiC. Indeed in the magnetic field dependence, the ODMR and PDMR both show a 70 MHz ZFS and a g-factor with very low spin-orbit contribution. Also, the Rabi frequency of the PDMR are very similar in frequency to the ODMR of the V_{Si^-} , indicating both detected signals arise from a Spin 3/2 system. These together prove that indeed it is a V_{Si^-} defect contribution. We explained this on page 11, line 2 starting with "To check that the measured signal originates from V_2 centers, we measure the ground state ZFS via Zeeman splitting measurements by observing the resonances at various magnetic fields...."

The reviewer is right in pointing out that larger magnetic fields might be better suited for measuring a g-factor. However, the clear attribution is done by the ZFS, and thus smaller field values are better suited. Also, in our case magnetic field can only be provided up to about 70 G with well controllable magnetic field strength and orientation. As the RF frequency is determined by the ZFS, g-factor and magnetic field, the frequency range is fixed up to about 300 MHz for such measurements.

Comment 2-d: 4) I understand that the Ni layer in the used samples is needed to produce a Schottky contact and also, as explained in the SI, it is needed as etch stop during the sample preparation. I wonder though whether there could be other, non-ferromagnetic materials that could work in a similar way? If not, I would like the manuscript to be more explicit in the main text about why the sample has to be designed that way it is..

Answer: Indeed, in general other contact materials such as Ti, Al, Mo, W are available. Also, graphene as a contact material is very interesting for this application. We are planning on using such materials in further studies, but here Ni was used as it is the standard material for n-type contacts on 4H-SiC with well-known processing steps [Advancing Silicon Carbide Electronics Technology I, page 71ff].

Action taken: Added a sentence on graphene as alternative contact material (changelog M26).

Comment 2-e: 5) When it comes to figures, I would shift panel (d) in Fig. 1 into Fig. 3 as the experiment and the chosen readout sequence are not really discussed before.

Answer: We thank the reviewer for this suggestion. As we added a section discussing the experimental setup and pulse sequences in the Supplementary Methods 1, we removed panel (d) from Fig. 1 and reference the Supplementary Information instead.

Action taken: Removed panel Fig. 1(d), added detailed explanation of pulse sequences to Supporting Information (changelog S2, M11-M12, M18).

Comment 2-f: 6) In figure 3, I would not use dashed and dotted lines. The fit plots are hardly distinguishable from the data points.

Answer: We agree that overlapping data points and lines are difficult to distinguish. We feel the new style improves on this.

Action taken: Changed the plot style for better readability (changelog M2-M4).

Comment 2-g: 7) I would have appreciated a discussion about the observed coherence times of the resonantly excited and electrically read spin states. Rabi oscillation, of course, does not provide exact information about T₂ spin relaxation times. However, it does allow for an estimate and a discussion of T₂, i.e. a discussion of what interaction of the paramagnetic defect could potentially be coherence time limiting. For instance, from the measurement of the random magnetic field distribution as shown in Fig. S8 of the SI, one may or may not be able to determine whether the observed Rabi oscillation dephasing is due to coherent dephasing throughout the ensemble of VSi- centers or whether it actual is due to real coherence decay. If the former turned out to be the case, the electrically detectable spin coherence times could even be longer than the microsecond scale which they appear to have and that would make the outcome of this study even more significant.

Answer: The Reviewer is correct, the Rabi decay does not give a proper indication of T₂ times. As stated in the manuscript, the Rabi decay timescale by two methods are on a similar scale (exponential decay fit: ODMR: 2.2±0.5 μs, PDMR: 1.1±0.1μs). In the beginning, we were not able to detect Hahn-Echo. In the meantime, we achieved better signal-to-noise ratio by changing the ratio of averaging and magnetic field settling time. We can now report on Hahn-Echo measured by ODMR and PDMR and added the newly acquired T₂ data with detailed explanations of what can be stated by them into Supplementary Information.

Action taken: Measured T₂ times, added section “7 Hahn-Echo measurements” with discussion to the SI (changelog S4, S22, M25).

Answers to Reviewer #3:

Comment 3-a: [...] The authors present compelling data for the PDMR detection but offer little in the way of understanding regarding the physical underlying mechanism.

Specifically –

i) The discussion of the mechanism behind the current increase and decrease (page 4-5 of main text) is poorly explained. A discussion beyond just simply acknowledging the competing rates would give better intuition into the technique.

Answer: We thank the reviewer for pointing out that this section is hard to follow. We agree that a thorough discussion would be interesting. However, our current setup does not allow measurements to correctly acquire all rates involved (e.g. due to ensemble statistics, unknown recharging process, number of defects and surrounding traps, valence & conduction band contribution). At the moment, we do not see a feasibility to model the process confidently, as there are too many free parameters involved. Especially as we do not see a simple increase or decrease in current as expected, but a complex behaviour depending on position and focusing, we figure that the surrounding defect environment has a crucial impact on this behaviour and so far cannot be predicted. For these reasons, we did not provide further discussions about the competing rates.

Action taken: Revised PDMR explanation (changelog M27-M29).

Comment 3-b: ii) The discussion regarding the spatial dependence of the PDMR signal and the relative position of the optical spot (Figure 3d) is not well explained. What are the possible underlying mechanism behind this PDMR spatial dependence?

Answer: At the moment, the observation is, that this can be a function of position and focusing depth. The signal seems to depend on the defects and the defect/trap environment surrounding the focal spot and the present Fermi level. Also, a change in band-bending upon light illumination might be contributing [Nat. Electron 1, 502–507 (2018)] (see answer to **Comment 2-b**). Although the observed spatial dependence is very interesting, we want to focus on improving SNR of the used PDMR scheme in future, that will allow to systematically investigate our observations.

Action taken: Added paragraph on spatial dependence of PDMR signal (changelog M24).

Comment 3-c: iii) Please provide better intuition regarding the surprising change in sign of the PDMR signal (page 11). What are the possible underlying physical mechanisms that contribute to this sign change? What surrounding defects might affect this, and how might these effects be mitigated?

Answer: As the Fermi level can easily be pinned by the V_C , we think that reducing the density of V_C is the most interesting point for further looking into this effect. This can be done using special growth methods [J. Cryst. Growth 281, 370 (2005)] and post processing [Appl. Phys. Express 2, 41101 (2009)]. Also see answers to **Comment 3-b** and **Comment 2-b**.

Action taken: Added paragraph on spatial dependence of PDMR signal (changelog M24).

Comment 3-d: After successfully addressing a few of these comments, publication in Nature Communications will be recommended from my side.

Answer: We appreciate the referees comments which helped us to improve the manuscript as advised by the referee. We believe that a publication can raise interest of readers and initiate discussion and further investigation of the underlying processes.

Comment 3-e: I also have a number of minor issues/comments which are intended to make the presentation clearer and easier to follow:

a) Related to point i) above, in the main paragraph on page 4 starting with ‘In the following, ...’, there is a lot of detailed description about the underlying charge dynamics but the discussion is worded and formatted in a confusing fashion. While the paragraph does refer to figures 1(a) – (c) as a whole, a proper breakdown and description of each of the different possible photo-ionization mechanisms will bring clarity.

Answer: We thank for pointing this out. As also advised in **Comment 3-a** and **Comment 3-f**, we have revised this whole paragraph.

Action taken: Revised PDMR explanation (changelog M27-M29).

Comment 3-f: b) The discussion regarding the ISC rates (page 4) could use a more detailed discussion, possibly in the SI.

Answer: We believe that this discussion is important. We extended the explanation in the main text for a better understanding by the readers. See comment **Comment 3-a** for details.

Action taken: Revised PDMR explanation (changelog M27-M29).

Comment 3-g: c) When referring to the supporting information, you could also include specific sections for clarify and convenience.

Answer: We thank the reviewer for pointing out this issue. We agree that more clear references to and sectioning of the supporting information improve readability and implemented changes accordingly.

Action taken: Added section numbering and corresponding references to Supporting Information (changelog S1, M8-M10).

Comment 3-h: d) How does the reduced contact quality help minimize the dark current? (Page 6)

Answer: As the internal resistance of the device increases, the collection efficiency for electrons drops. While this is detrimental to the PDMR signal, the linear photocurrent contribution as well as leakage dark currents through the device are smaller. This allows for larger transimpedance gain which in turn allows for a better SNR in transimpedance detection for small currents. The reviewer is correct, as this is more precisely described as leakage current.

Action taken: Changed the word from “dark current” to “leakage current” to avoid confusion (changelog M1).

Comment 3-i: e) What is the motivation behind the dimensions of the n⁺⁺/n-n⁺⁺ structure? (Page 6) Why did you use this type of doped structure?

Answer: Structures for photocurrent detection are usually available as Schottky type, p-i-n or p-n junctions. Here, we used a metal-semiconductor-metal type device, as it does not need p-type doping. This would have required either ion implantation doping (introducing other defects) for lateral devices or a more complex device geometry, as optical access to the diode structure is more difficult to achieve in vertical diodes. Note that the detected very small currents can be further diminished by the high resistivity of e.g. ITO.

The n⁺⁺ structure is needed to form better contact to the metallization and block the depletion layer from the Schottky contact not to affect the Fermi level of the n- region. The n-region itself fixes the Fermi level so that V_{Si} is in the right charge state and contains the optically active volume. Unfortunately, diodes also act as detectors for RF and MW signals. We thus wanted to minimize device footprint and capacity to minimize RF coupling and thus did not use classical interleaved finger structures. All together, this allowed us to create a lateral photodiode structure of small size that still allows for optical access and requires neither implantation nor complex fabrication techniques.

The device behaviour then is largely influenced by the electron irradiation, but we settled for values with which we achieved good results in optical experiments in prior work. Overall, the structure shown in the manuscript proved to provide the required characteristics to detect the signal. The optimization of device structure and type clearly is one of our goals for the future and seems feasible as now a proof-of-principle is demonstrated.

Action taken: Clarified statement on device structure and RF coupling issues (changelog M5).

Comment 3-k: f) Is there a specific reason why 429 Hz was chosen as a lock-in frequency? (Page 8)

Answer: The -3dB-bandwidth-limit of the optical detector is 750Hz and the transimpedance amplifier is limited to 1kHz. Thus we have chosen a frequency, which is well below this limit and not a multiple of power line frequency of 50 Hz. While higher frequencies in general reduce 1/f voltage-noise, they tend to add current-noise in the trans-impedance case, which can be seen in transimpedance amplifier application notes [NOISE ANALYSIS OF FET]

TRANSIMPEDANCE AMPLIFIERS]. This means, that an optimal frequency point can be found. An integrated chip very similar in terms of specification to the amplifier used in our measurements, reveals that for a [Analog Devices ADA4530-1], the current noise density is more detrimental than the $1/f$ noise after 1Hz at the amplifications used (gain settings: $1e8$, $1e9$).

We then chose a lock-in frequency in between the $1/f$ corner frequency, which is still far enough from the -3dB bandwidth in order to not observe phase and amplitude changes by the amplifier's transfer function.

In the future, scanning the lockin-frequency for the minimum noise region might be an additional suitable approach to further reduce noise. However, as at the moment the stray contribution of applied RF fields is strong, at the current state this is not a major noise source to consider.

Action taken: Added section Supplementary Methods 1 to Supporting Information with short explanation (changelog S2).

Comment 3-m: g) The phrase 'The beam in front of the photodiode is attenuated by an iris to prevent detector saturation' (page 8) is unclear and out of place. Firstly, this feels like a detail that could be described in better in the supplemental, and further motivated as to the use of an iris (as opposed to a neutral density filter) and what sort of attenuation is expected and needed.

Answer: We thank the reviewer for pointing out this issue. The lack of readability of the experimental setup has been addressed with an experimental setup section in the supporting information.

A variable iris was used instead of an ND filter simply because of availability at that moment. In this specific setting, we did not see any advantage or disadvantage of an iris over a neutral density filter.

Action taken: Section with description and sketch of the experimental setup added as Supplementary Methods 1, removed the sentence from main text (changelog S2, M20).

Comment 3-n: h) You mention 'the magnetic field is roughly aligned along the c-axis' (Page 8). How susceptible is this effect to field misalignment in general? Do you attribute any PDMR effect to changes in the B-field alignment during the sweep?

Answer: Overall, the B-field alignment is important for the presented measurements, but not crucial. As can be seen in publications [Phys. Rev. B 92, 115201 (2015)] on magnetic field sensing with the silicon vacancy, miss-alignments of up to 10 degree against the c-axis do not result in major degradation of signal contrast when the B-field strength is larger than the zero-field splitting. The splitting between the two visible transitions will always decrease from its theoretical maximum of 140 MHz. As can be seen in the Zeeman-splitting dependence, the alignment of the B-field to the c-axis was within the tolerance and the splitting is in the range of 139-140 MHz (approx. 4 degree).

The magnetic field itself is assumed to be stable, as electromagnets with constant current control in Helmholtz configuration are used. By this, small drifts of the sample also do not impact the magnetic field. However, during ramping the magnetic field B_z component, small misalignments can happen due to stray magnetic fields. As the field magnitudes used are larger than the residual fields of a few G in our setup, we do expect any relevant impact on the data presented. Thus, during the measurement times, we do not assume any effective change of the B-field alignment. Mainly the local variation in B-field due to the Ni contact ferromagnetic layers seem to be more crucial in combination with drift of sample and stage.

Comment 3-o: j) My understanding is the off-resonant signal is obtained as a single magnetic field (corresponding to 23 MHz). Why was this not done at all steps of B-fields?

Answer: The subtraction of background was only performed at measurements of Rabi oscillations and the mapping of the PDMR amplitudes. These types of measurements contain only one frequency-magnetic-field pair corresponding to an on-resonant condition. Thus, a single off-resonant frequency-magnetic-field-pair was used to obtain the background signal and no further B-field steps are used in these measurements. We used this technique to extract the real PDMR signal amplitude.

For the pulsed PDMR measurement, this subtraction was not necessary. This is due to the fact that in this case different frequency-field pairs are used for every datapoint and result in a background signal. Thus, the PDMR amplitude is available as fit parameter of the PDMR peak.

Action taken: Added pulse sequences with visible B_0 behavior for clarity (changelog S2).

Comment 3-p: k) What is the estimated detection volume? (page 9) You reference the sample geometry but is the volume assumed to be the wide of the gap and the thickness of the n- or a subset thereof?

Answer: The detection volume differs for the ODMR and PDMR case. In the ODMR case, the volume is approximately given by the overlap of optical excitation and optical detection volume. This usually is given as confocal volume, which usually assumes the diffraction-limited resolution of the microscope. As no spatial filtering was used, the detection volume is larger than this expectation. For the PDMR, the volume is the two-photon excitation volume, which is calculated using the square of a gaussian beam intensity distribution. The detection volumes and especially the number of addressed defects change more than two orders of magnitude on slight parameter changes (e.g. objective back-aperture fill factor), which we consider not profound enough. Additionally, as Fig. S4. shows, the fully focused position does not lead to a measurable PDMR signal, as it seems not enough defects contribute to the signal in this smallest detection volume. Defocusing adds another parameter of uncertainty, unfortunately rendering such approximated numbers invalid.

Comment 3-q: l) Regarding the phrase 'likely generated by excitation of other intra-band defects ...' (Page 13) - Are there any indications what those other defects might be? Can specific defects be attributed to the spatial dependence or the sign change observed in the PDMR signal?

Answer: The major abundant defects we expect in the sample are the following:

- 1.) V_S : this is created homogeneously (cubic and hexagonal sites) by the electron irradiation and is the defect we addressed.
- 2.) V_C : The carbon vacancies are created during electron irradiation as well and are known to be a major source for reduction of minority carrier diffusion length in this material [Appl. Phys. Lett. 88, 052110 (2006), Appl. Phys. Lett. 90, 202109 (2007)]. These defects are also expected to be homogeneous throughout the sample. It can exist in various charge states. The initial doping condition suggests a neutral charge state, rendering a single-photon absorption to the conduction band possible with wavelengths shorter than 850 nm [Phys. Rev. B 98, 195202 (2018)], J. Appl. Phys. 119, 235703 (2016)]. However, as so many defects are introduced by electron irradiation, the doping is expected to be compensated and defects [Phys. Rev. B 75, 155204 (2007)] in the +1 charge state are expected to exist. This would lead to a

single-photon absorption for lasers with 830nm [J. Appl. Phys. 119, 235703 (2016)], and thus contribution to the linear photocurrent might be well possible with a 785nm laser even under compensated conditions.

- 3.) N: As nitrogen donors are created during the growth process for doping, they provide shallow states close to the conduction band. [Phys. Rev. B 71, 241201 (2005)]. However, they should be compensated after irradiation and we expect their contribution to photocurrent to be negligible.

Note that although other defects can contribute to the photocurrent, the measurement is selective to V_{Si} due to locking detection of the changes in photocurrent at the resonant MW modulation. We cannot attribute the sign change to a specific defect. However we believe it is most likely that charge interplay of V_{Si} , V_C and other traps combined with band-bending and illumination create this effect [arXiv:1906.05964].

Action taken: Added paragraph on most likely defects for linear photocurrent contribution (changelog M30).

Comment 3-r: m) The discussion regarding the detection volume and leakage currents (page 13) could be expanded. What currently limits the detection volume? What is the estimate 'small area within the aperture' that shows PDMR contribution? I understand this might be a complex process but is there a possible intuitive picture that might be discussed or is this not well understood?

Answer: As this is the first study on PDMR in SiC, we do not fully understand the effect of the device design and the other competing process so far. An intuitive idea is that the band-bending in the device and local Fermi level pinning by other defects provide the correct defect charge only locally (also see answer to **Comment 2-b**). At the moment, the maximum excitation volume (and therefore the detection volume) is limited by the maximum laser power density as this is primarily a two photon process. The reviewer is right, in that this sentence is unclear and we improved the explanation.

Action taken: Improved explanation (changelog M23,M24).

Comment 3-s: Supplemental:

n) Please add section markings to each section and refer specific discussions in the main text to the requisite section. This makes things far easier to follow

Answer: We agree that proper sectioning and references improve the readers experience.

Action taken: Added section numbering and corresponding references to Supplementary Informations (changelog S1, M8-M10).

Comment 3-t: o) What is the intent of the post-irradiation I-V characteristic plot? (Figure S2b) Is the intent to show the effect of irradiation-induced defect? This would be a good place to discuss what other possible defects might contribute.

Answer: The reviewer is correct, the plot was placed to show the impact of high dose electron irradiation on the sample. Indeed, the possible contributing defects can be discussed more thoroughly at this place.

Action taken: Added a paragraph with details on carbon vacancy (changelog S5,S6).

Comment 3-u: p) In the SI (S-5) you detail a focal dependence of the PDMR signal inside the device. Expanding on this sort of discussion would be useful in addressing general comment (ii) as well as the comment on the estimated detection volume (k) and (m). How

does the small excitation volume relate to the detection volume? What would you estimate the minimum illuminated defect density you would need to find PDMR signal?

Answer: As stated in answer to **Comment 3-p**, the relation between the detection volume can be calculated theoretically, but leads to overly sensitive dependence on unknown parameters, rendering their usage invalid. Also, for the current per defect, internal amplification of the device might be considered. As our SiC device is highly electron irradiated, the electric properties are strongly modified to the former well-defined state. PDMR from single NV centers has been detected. In principle, we think this is feasible also for SiC. This requires much better understanding of the limitations, better control over the surrounding defect density and refined devices and fabrication. If we assume that we saturate the photoionization rate (6ns ES lifetime => 25pA) , require at least one ISC transition for spin polarization (100ns => 1.5pA) and use spin-control RF pi-pulses (300ns), we roughly estimate a PDMR amplitude of 400 fA per defect. This theoretically is measurable, however neglects electron collection efficiency, amplification and many other effects.

Comment 3-v: q) The discussion related to the SNR ratio is confusing (Page S-9). Specifically, the discussion related to the lock-in constant offset and the various definitions of 'contrast' is not clearly detailed.

Answer: We agree that, the discussion is quite complicated to follow as is and rewriting of this specific part is advisable. We want to emphasize the importance of this section to the field, as usually, contrast definition is deducted from the NV centers in diamond, which is not applicable for other defects or techniques. This is a long standing tradition, that we feel with upcoming other defects, materials and measurement techniques (like PDMR, that can change sign also for NV centers between measurements) now imposes problems in scientific communication of signal strength and results in misleading numbers, which are difficult to compare in a proper way. This is why we feel obliged to introduce this more general definition of signal contrast, which similarly also is used in many other fields.

Action taken: Shortened and revised paragraph (changelog S10-S14).

REVIEWERS' COMMENTS:

Reviewer #1 (Remarks to the Author):

Authors have answered the previous comments properly and I can now recommend its publications.

Reviewer #2 (Remarks to the Author):

The revised manuscript NCOMMS-19-11805A, "Coherent electrical readout of defect spins in silicon carbide by photo-ionization at ambient conditions" by Niethammer et al. has been prepared in response to the three reviews. Overall, I have focused this second review on the specific remarks made by all three reviewers and I find that most remarks (not just my own) about the original manuscript have been appropriate. I find that the authors have looked carefully at the remarks and responded adequately, i.e. by carefully implementing the advice wherever possible. In my view, this has made the manuscript overall much more readable and the Supporting Information document better structured and informative. For me, there are no remaining open questions. I, therefore, recommend publication of this second manuscript iteration. *** end of report ***

Reviewer #3 (Remarks to the Author):

The authors have responded thoughtfully to the comments raised in the previous report. While they were unable to directly address all the comments due to experimental limitations and complex dynamics, their discussions both in the response and main text were sufficient to provide added contextual understanding. I appreciate the rewriting of the PDMR mechanisms section and the addition of the pulse sequence diagrams in the supplemental. At this point I believe the paper is substantially clearer and would recommend publication in Nature Communications.

I have two minor suggestions:

In response to comment 3-i:

I believe the motivation behind the electronic device structure is useful and merits being included in the text. The response to this comment was well formulated and could be a good addition to the supplemental methods (in Section 2)

In response to comment 3-p:

I feel a direct comparison between ODMR and PDMR is still difficult without understanding of the estimated detection volume of each technique. I understand the methodology and technical difficulties associated with estimating such a volume and appreciate the added discussion related to the contributing volume in the main text (page 13). While that discussion provides some intuition, I would still like to also see a quantitative discussion of the geometric limits to the detection volume estimated in the supplemental information.

Point-to-point answers to the reviewers

Reviewer #1 (Remarks to the Author):

COMMENT 1-a: Authors have answered the previous comments properly and I can now recommend its publications.

ANSWER: We thank the reviewer for his or her positive feedback.

Reviewer #2 (Remarks to the Author):

COMMENT 2-a: The revised manuscript NCOMMS-19-11805A, “Coherent electrical readout of defect spins in silicon carbide by photo-ionization at ambient conditions” by Niethammer et al. has been prepared in response to the three reviews. Overall, I have focused this second review on the specific remarks made by all three reviewers and I find that most remarks (not just my own) about the original manuscript have been appropriate. I find that the authors have looked carefully at the remarks and responded adequately, i.e. by carefully implementing the advice wherever possible. In my view, this has made the manuscript overall much more readable and the Supporting Information document better structured and informative. For me, there are no remaining open questions. I, therefore, recommend publication of this second manuscript iteration.

ANSWER: We thank all reviewers for their valuable comments and suggestions, which helped a lot in improving the manuscript.

Reviewer #3 (Remarks to the Author):

COMMENT 3-a: The authors have responded thoughtfully to the comments raised in the previous report. While they were unable to directly address all the comments due to experimental limitations and complex dynamics, their discussions both in the response and main text were sufficient to provide added contextual understanding. I appreciate the rewriting of the PDMR mechanisms section and the addition of the pulse sequence diagrams in the supplemental. At this point I believe the paper is substantially clearer and would recommend publication in Nature Communications.

ANSWER: We are pleased to hear that understanding has improved.

COMMENT 3-b: In response to comment 3-i:

I believe the motivation behind the electronic device structure is useful and merits being included in the text. The response to this comment was well formulated and could be a good addition to the supplemental methods (in Section 2)

ANSWER: We agree that this information can help in reproducing this work and improving future devices. We thus added a paragraph to the Supplementary Information (changelog S8).

COMMENT 3-c: In response to comment 3-p:

I feel a direct comparison between ODMR and PDMR is still difficult without understanding of the estimated detection volume of each technique. I understand the methodology and technical difficulties associated with estimating such a volume and appreciate the added discussion related to the contributing volume in the main text (page 13). While that

discussion provides some intuition, I would still like to also see a quantitative discussion of the geometric limits to the detection volume estimated in the supplemental information.

ANSWER: We agree that the comparison of SNR between the two methods is not normalized to the number of contributing defects so far and the single spin detection is highly desirable. As stated previously, we are not confident in our quantitative estimations of the detection volume and hope to find better experimental methods to access this information in a future work more precisely. However, we agree in the reviewer's strong argument of comparability between ODMR and PDMR and thus decided to add a section with rough estimation for the detection volume and number of defects to the Supplementary Information (changelog S10, M20).